# Are workplace factors associated with employee alcohol use? The WIRUS cross-sectional study

Mikkel Magnus Thørrisen ![ORCID],[1,2] Jens Christoffer Skogen,[3,4] Tore Bonsaksen,[5,6] Lisebet Skeie Skarpaas,[1] Randi Wågø Aas[1,2]

For numbered affiliations see end of article.

**Correspondence to**
Dr Mikkel Magnus Thørrisen;
mikkel-magnus.thorrisen@
oslomet.no

## ABSTRACT

**Objectives** Sociodemographic predictors of employee alcohol use are well established in the literature, but knowledge about associations between workplace factors and alcohol use is less explored. The aim of this study was to explore whether workplace factors were associated with employee alcohol use (consumption and alcohol-related problems).

**Design** Cross-sectional study. Linear and binary logistic regression analyses.

**Setting** Heterogeneous sample of employees (workers and supervisors) from 22 companies across geographical locations and work divisions in Norway.

**Participants** Employees (N=5388) responded on survey items measuring workplace factors and alcohol use.

**Outcomes** Data on alcohol use were collected with the Alcohol Use Disorders Identification Test (AUDIT). Consumption was measured with the AUDIT-C (the first three items), and alcohol-related problems were operationalised as a sum score of 8 or higher on the full 10-item AUDIT.

**Results** Higher levels of alcohol consumption were associated with more liberal workplace drinking social norms (b=1.37, p<0.001), working full-time (b=0.18, p<0.001), working from holiday home (b=0.40, p<0.01), being a supervisor (b=0.25, p<0.001), having supervisors with less desired leadership qualities (b=−0.10, p<0.01), shorter working hours (b=−0.03, p<0.05), higher workplace social support (b=0.13, p<0.05) and higher income (b=0.02, p<0.001). Alcohol-related problems were associated with more liberal workplace drinking social norms (OR=3.52, p<0.001) and shorter working hours (OR=0.94, p<0.05).

**Conclusions** Workplace drinking social norms were the supremely most dominant predictor of both consumption and alcohol-related problems. Results suggest that some workplace factors may play a role in explaining employee alcohol consumption, although the predictive ability of these factors was limited. This study points to the importance of drinking social norms, workplace drinking culture and leadership for understanding employee alcohol use.

## STRENGTHS AND LIMITATIONS OF THIS STUDY

⇒ This study explores the importance of a wide range of workplace factors for employee alcohol use in a large and heterogeneous sample by means of a validated alcohol screening instrument.

⇒ This study uses individuals' overall degree of consumption, as well as odds of having a drinking pattern that may induce alcohol-related problems, as outcomes.

⇒ Study limitations include a low response rate, the use of self-reported data and a cross-sectional design that precludes causal inferences.

## INTRODUCTION

Alcohol consumption is associated with detrimental health outcomes.[1–4] Globally, approximately 5% of mortality and disability-adjusted life years (DALYs) can be attributed to alcohol consumption.[5] Within the age group of 25–49 years, 1.6% of global DALYs were attributable to alcohol use disorders in 2019.[6] Risky drinking has been conceptualised as a drinking pattern that increases the risk of social, legal, medical, occupational, domestic and economic problems.[7] Reducing harmful alcohol use represents a keystone in sustainable development of health.[5 8]

Alcohol is the most commonly used and misused psychoactive substance in the workforce.[9] Between 10% and 30% of employees consume alcohol at a risky level and may benefit from alcohol prevention interventions.[10] Employees' alcohol use is associated with increased sickness absence[11–14] and presenteeism (impaired on-the-job performance).[15 16] Research has demonstrated that brief alcohol interventions have promising effects in primary care,[17 18] as well as in working populations.[19–21]

According to Frone's model of employee substance use,[22] employee alcohol use can be construed as a function of individual and workplace environment characteristics, and multifaceted interactions between them. Individual (sociodemographic) predictors of alcohol use among employees are quite well established in the research literature. For instance, being male, young, unmarried

and not having children constitute individual predictors of elevated alcohol consumption.[10 23 24] The role of workplace factors is, however, not as established in the literature. Workplace factors may, on the one hand, refer to employment characteristics, that is, to more general workplace factors conceptualised in the form of conditions and terms defining the work situation (eg, working hours, work schedule, job position, job size and income). On the other hand, the workplace can be understood as a psychosociocultural setting that involves job demands and support factors (which are related to occupational stress, for example, psychological job demands, job control, workplace social support and supervisors' perceived leadership qualities), travels and worksite factors (related to degree of workplace social control, for example, job travels, working from home and working from holiday home) and workplace drinking social norms (which reflect the workplace alcohol use climate, for example, workplace drinking culture and drinking social norms).

Studies have demonstrated that workplace alcohol use climate may affect the drinking level of employees,[25–27] for example, through the impact of injunctive and descriptive norms.[28] 'Drinking social norms shared by a group define standards of appropriate behaviour, creating social controls that regulate workplace alcohol availability and drinking behaviours'[26] (p. 602). For instance, in a large-scale study of employees in the USA (across workgroups and worksites), it was found that being part of an alcohol-discouraging workplace drinking culture predicted lower consumption (less frequent drinking in general and at work, as well as overall reduced likelihood of being a heavy drinker), compared with being part of a more encouraging workplace drinking culture.[26] Using multilevel analysis, it has been found that individual-level measurement of drinking social norms may be indicative of group-level norms at worksites employees are nested within.[26] Research on the importance of other aspects of the workplace and work situation, such as general employment characteristics, job demands and support factors and travels and worksite factors, stands out as more inconclusive.

Systematic reviews generally point to longer working hours being associated with higher alcohol consumption.[29 30] However, the evidence for such a relationship remains uncertain. Some studies have failed to demonstrate statistically significant relationships,[31–33] and an opposite association (shorter working hours associated with higher risk of alcohol dependence) was found in a study of employees in Taiwan.[34] In a Norwegian study,[35] having flexible working hours was associated with increased consumption, and some studies have linked shift work with increased probability of workplace alcohol use,[31] as well as with higher overall consumption.[36] Others have not found a relationship between shift work and alcohol use.[35 37–39] Evidence on the importance of work position is mixed. Some studies support an association between being a supervisor (as opposed to a worker) and higher alcohol consumption,[32 40] while others do

not.[23 31 34 35] In a study of employees in the USA, Lee *et al* revealed that working part-time (as opposed to full-time) was associated with higher alcohol use per week, but unrelated to prevalence of binge drinking episodes and alcohol-related consequences.[41] Although working hours may be somewhat overlapping with job size, one may capture different aspects of 'time at work' by studying typical number of hours at work per day (working hours) and whether or not an individual has a full-time job (job size). For instance, variations in working hours that exceed 'normal working hours' (eg, overtime) will not be captured in a dichotomy of part-time and full-time work. Studies have suggested that higher socioeconomic status is associated with higher levels of alcohol consumption,[42 43] which may be due to alcohol being a costly commodity.[44] As such, high-income employees may have better access to alcohol than low-income employees. A study of Canadian employees found that employees in high-income households had 35% higher odds of high-risk drinking than employees in low-income households.[23] However, in a study of Japanese employees,[45] family income was unrelated to prevalence of alcohol-related problems.

Travels and worksite factors may refer to the degree to which employees are subjects of direct monitoring at the workplace. According to Roman and Trice,[46] low visibility at the workplace and low degrees of interdependence among employees stand out as risk factors for deviant drinking behaviours. In a study of employees in Norway,[40] working from home (telecommuting) at least 2 days per week was associated with higher alcohol use. Furthermore, a high prevalence of job travels has been linked with increased likelihood of problematic drinking.[47]

Research on the importance of job demands and support factors (eg, leadership, job demands and control, and workplace social support) for employee alcohol use has yielded variable results. It has been advocated that visible and active supervisors constitute a protective factor for alcohol use among workers,[46] while others have failed to demonstrate such a relationship.[47 48] In a Norwegian study,[49] supervisors' leadership style (transformational and laissez-faire leadership) was not related to employee alcohol use. On the other hand, some studies imply that alcohol use and hangover episodes are less prevalent at workplaces where supervisors exert active social control.[25 50 51] Analyses of cross-sectional data imply that lower levels of job control (skill discretion and decision latitude) may be associated with higher levels of alcohol consumption, although such a relationship has proven difficult to replicate in longitudinal data.[52] Some studies reject a potential association between job control and alcohol use,[23 34 53] while others underscore that different types of control may be differentially related to consumption. For instance, Marchand found that an increased probability of risky drinking was associated with lower skill discretion and higher decision latitude,[32] while Saade and Marchand found the opposite.[33] Higher perceived psychological job demands have in some studies been related to higher alcohol use,[33 34]

but not in others.[54] In a study of Canadian employees,[55] lower psychological demands were associated with higher alcohol use for men only. The evidence base of physical job demands is also inconclusive. A Canadian study found that higher physical demands were associated with higher alcohol use among men only,[55] while a French study found an association for women only.[54] Another Canadian study failed to demonstrate a relationship between physical job demands and alcohol use altogether.[33] Some studies indicate that lower workplace social support is associated with higher alcohol consumption,[33 38 55] while others do not.[23 53] A study from Norway found increased odds of alcohol-related problems among employees who reported a perceived imbalance between work efforts and rewards, in conjunction with high levels of work overcommitment.[56]

In summary, evidence on associations between workplace factors and employee alcohol use stands out as complex and nuanced. Moreover, few studies have explored the importance of workplace factors for alcohol consumption as well as for odds of alcohol-related problems in heterogeneous employee samples by means of validated alcohol screening instruments. Individuals' overall degree of consumption, and their odds of having a drinking pattern that may induce alcohol-related problems, constitute two different—yet equally important—ways of conceptualising alcohol use that can be differentially associated with workplace factors. Knowledge about associations between workplace factors and employee alcohol use is important as an aid in determining the extent to which employers should make workplace-based alcohol health promotion and prevention interventions an overall priority, and for determining which elements should be emphasised in such interventions. Hence, further research is warranted.

## Study aim
The aim of this study was to explore whether workplace factors (employment characteristics, job demands and support factors, travels and worksite factors and workplace drinking social norms) were associated with employee alcohol consumption and odds of having alcohol-related problems in a heterogeneous sample of employees in Norway.

## MATERIALS AND METHODS
### Design and setting
This cross-sectional study explored associations between workplace factors and alcohol use in a heterogeneous sample of employees (workers and supervisors) in 22 companies in Norway (private companies: n=8; central government companies: n=6; local government companies: n=8), across geographical locations. The study was part of the Norwegian national WIRUS Project (Workplace Interventions preventing Risky alcohol Use and Sick leave).

### Data collection and sample
The researchers recruited companies through three occupational health service units' client lists in collaboration with the addiction competence environment KoRus Stavanger. Individual-level inclusion criteria were: (1) age 16–72 years; (2) status as employee, regardless of sector, work division or geographical location; and (3) basic understanding of the Norwegian language. Our definition of employees (criteria 2) included white-collar (supervisors, (semi)professional roles, business owners), blue-collar (labourers and skilled trade roles) and pink-collar employed (hospitality, retail, care and administration roles).[57] Inclusion of employees aged 16–72 years (criteria 1) was chosen because 16 constitutes the age when individuals in Norway have typically completed mandatory primary and secondary school and are eligible for full-time work (or further education), and because employment by Norwegian law may be terminated by the employer at age 72 years (even though 67 years constitutes the general age of retirement).[58] Although Norwegian legislation forbids retail, serving and supply of alcohol to anyone under the age of 18 years,[59] alcohol is regularly consumed by adolescents.[60] Hence, it was deemed appropriate to include respondents aged 16 and 17 years in the study. The invited sample was, in accordance with the inclusion criteria, heterogeneous, representing different sectors, work divisions and geographical locations. Twenty-two companies agreed to participate and provided email addresses for all their employees (n=30 811). Digital questionnaires measuring workplace factors, alcohol use and sociodemographic variables were distributed to all employees in the 22 companies. Consent to participate was provided by 8542 employees (response rate=27.7%). A total of 3154 employees failed to respond on all relevant study items, leaving a final study sample of 5388 employees. Characteristics of the study sample are presented in table 1.

Data from participants were collected between October 2015 and October 2019. Hence, each participant was measured once (between 2015 and 2019), and all data were collected prior to the COVID-19 pandemic. We predefined that a statistically satisfactory sample size had to conform with the formula N>50+(8×number of predictors/covariates),[61] as well as exceeding a ratio of 15 participants per predictor/covariate variable.[62]

### Measures
#### Employee alcohol use (outcomes)
The outcomes were measured with the Alcohol Use Disorders Identification Test (AUDIT). The AUDIT is an alcohol screening instrument, developed by the WHO,[7 63] consisting of 10 items (each scored 0–4). Higher scores on the AUDIT indicate higher consumption and more severe alcohol-related consequences. The first outcome, *alcohol consumption*, was measured with the first three AUDIT items (sum score of three items=continuous scale 0–12), often referred to as the AUDIT-C.[64] The second outcome, *alcohol-related problems*, was measured as a dichotomous

**Table 1** Characteristics of the study sample

| Variable | |
|---|---|
| Gender | |
| Male, n (%) | 1699 (31.5) |
| Female, n (%) | 3689 (68.5) |
| Age (years), M (SD) | 44.8 (11.2) |
| Educational attainment | |
| Primary/lower secondary, n (%) | 87 (1.6) |
| Upper secondary, n (%) | 1006 (18.7) |
| University/college ≤4 years, n (%) | 2125 (39.4) |
| University/college >4 years, n (%) | 2170 (40.3) |
| Employment sector | |
| Private corporation, n (%) | 368 (6.8) |
| Local government, n (%) | 3469 (64.4) |
| Central government, n (%) | 1551 (28.8) |
| Work division | |
| Transportation/manufacturing, n (%) | 285 (5.3) |
| Public administration/services, n (%) | 4299 (79.8) |
| Health services, n (%) | 762 (14.1) |
| Other services, n (%) | 42 (0.8) |

n=5388.
M, mean; SD, Standard deviation.

categorical variable (0=no problems; 1=problems), based on a cut-off of 8 on the full 10-item AUDIT (scores 0–7=no problems; scores 8–40=problems).[7]

### Workplace factors (predictors)

Employment characteristics included *working hours* (number of hours during a typical day), *work schedule* (day job without weekends; day job with weekends; evening job; night job; shift work), *job position* (worker; supervisor), *job size* (part-time; full-time) and *income* (gross annual household income in Norwegian kroner).

Job demands and support factors included employees' perceptions of *psychological job demands, job control, workplace social support* and supervisors' *leadership qualities*. Psychological job demands (five items), job control (perceived control over workplace decisions and potential for using personal skills in the job; nine items) and workplace social support (from coworkers and supervisors; eight items) were measured with the Job Content Questionnaire (JCQ),[65] based on the Demand Control Support model,[66 67] each scored on 4-point Likert scales where higher scores indicated higher levels of demands, control and support, respectively. Reversed items were recoded and mean scores were calculated (demands: α=0.76; control: α=0.58; support: α=0.89). Supervisors' leadership qualities were measured with the Seven-item Leadership Qualities Questionnaire (7LQQ).[68] Developed on the basis of qualitative interview data,[69] 78 distinct leadership qualities were identified and categorised into seven desired leadership types (problem-solver;

contact-maker; responsible-maker; protector; trust-creator; recogniser; encourager), each type scored on a 5-point Likert scale based on the frequency of which supervisors were perceived to display the leadership type in question (0=never; 1=seldom; 2=sometimes; 3=often; 4=very often). A mean score of the seven items was calculated (α=0.94), where higher scores indicated higher levels of desired leadership qualities.

Travels and worksite factors (reflecting degree to which employees are under direct monitoring by coworkers and supervisors) were measured in terms of whether the job included *job travels* (no; yes), *working from home* (no; yes) and *working from holiday home* (no; yes).

Workplace alcohol use climate was conceptualised as employees' perceptions of *drinking social norms*, measured with the four items from the Drinking Norms Scale (DNS) that directly relates to the work situation.[26] Responses were provided on a 4-point Likert scale where higher scores indicated more liberal drinking social norms. A mean score was calculated (α=0.73).

An overview of workplace factors is presented in table 2.

### Covariates

The following sociodemographic factors, which have demonstrated significant associations with alcohol use in a Norwegian employee population,[10] were included as controls in adjusted analyses: *gender* (male; female), *age* (years), *educational attainment* (primary/lower secondary; upper secondary; university/college ≤4 years; university/college >4 years), *living status* (living alone; living with others), *marital status* (unmarried; married) and *number of children*. Due to the cross-sectional data in this study being collected over a period of approximately 5 years, *year of data collection* (2015; 2016; 2017; 2018; 2019) was included as an additional covariate.

### Analysis

Descriptive analyses were performed on the main study variables (outcomes and predictors) and presented in terms of means, standard deviations, frequencies and percentages, as appropriate.

Main analyses were conducted in two stages: model selection and predictor comparison.[70] In the model selection stage, associations between workplace factors and alcohol consumption (AUDIT-C scale 0–12) were explored by means of a multiple hierarchical linear regression analysis. Employment characteristics were entered as predictors in the first model, followed by sequential inclusion of job demands and support factors (model 2), travels and worksite factors (model 3), workplace drinking social norms (model 4) and covariates (model 5). Unstandardised (b) and standardised (β) regression coefficients were calculated. Potential multicollinearity was explored using the variance inflation factor (VIF). Multicollinearity was deemed a concern if VIFs exceeded 10.[71] Associations between workplace factors and the odds of alcohol-related problems (AUDIT-10 categories 0–7 vs 8–40) were explored with a multiple hierarchical

**Table 2** Overview of workplace factors

| Main factor | Factors | Measurement |
|---|---|---|
| Employment characteristics | Working hours | Continuous*: number of hours during a typical day |
| | Work schedule | Categorical*: day job, no weekends; day job with weekends; evening job; night job; shift work |
| | Job position | Categorical*: worker; supervisor |
| | Job size | Categorical*: part-time; full-time |
| | Income | Continuous*: gross annual household income in 100 000 NOK |
| Job demands and support factors | Psychological job demands | JCQ: mean score of five items |
| | Job control | JCQ: mean score of nine items |
| | Workplace social support | JCQ: mean score of eight items |
| | Supervisors' leadership qualities | 7LQQ: mean score of seven items |
| Travels and worksite factors | Job travels | Categorical*: no; yes |
| | Working from home | Categorical*: no; yes |
| | Working from holiday home | Categorical*: no; yes |
| Workplace drinking social norms | Drinking social norms | DNS: mean score of four items |

*Self-developed measure.
DNS, Drinking Norms Scale; JCQ, Job Content Questionnaire; 7LQQ, Seven-item Leadership Qualities Questionnaire; NOK, Norwegian kroner.

binary logistic regression analysis. Predictors and covariates were entered sequentially through five models, and odds ratios were calculated. Goodness of fit was explored by means of Hosmer and Lemeshow tests, where poor model fit was indicated by p<0.05.[72] In the final and fully adjusted models in the regression analyses (linear and logistic), associations between workplace factors and alcohol outcomes were adjusted for gender, age, educational attainment, living status, marital status, number of children and year of data collection. Work schedule (categorical nominal predictor) was dummy coded. All main analyses were performed with IBM SPSS V.27.

In the predictor comparison stage, the relative importance of predictors that demonstrated significant associations with the outcomes (p<0.05) in fully adjusted models in the model selection stage was estimated by means of dominance analysis (DA).[73] DA consists of examining $R^2$ values for all possible subset models in a multiple regression, with additional contribution of each predictor. A predictor is considered to be more important than its competitors if its predictive ability exceeds the others in all subset regressions[73] (p. 545). Each predictor's relative importance was expressed in terms of its average contribution (in percentage points) to the explained variance in the outcome. DA was conducted separately for each outcome using Stata V.17.

Significant results were defined as p<0.05.

### Patient and public involvement

Employee representatives and other relevant stakeholders were included in the WIRUS Project's reference group. Design, research questions, recruitment procedures and choice of outcome measures in the project have been informed by priorities, experience and preferences expressed by the reference group.

### RESULTS

Approximately 1 out of 10 employees (11.5%) reported alcohol-related problems (AUDIT ≥8). The sample's mean score on alcohol consumption was 3.2 (on the AUDIT-C scale ranging from 0 to 12). The majority of employees were workers (79.6% of the total sample) with full-time (81.8% of the total sample) day job schedules without weekends (73.9% of the total sample). Employees' average gross annual household income was 1 012 000 Norwegian Kroner (approximately £83 587), and a minority of employees had a job that included job travels (14.9% of the total sample), working from home (6.9% of the total sample) and working from holiday home (3.0% of the total sample). Job control and workplace social support were rated higher than psychological job demands (on the JCQ scales ranging from 1 to 4: $M_{Job\ control}$=3.0; $M_{Social\ support}$=3.2; $M_{Psychological\ job\ demands}$=2.2). Employees' mean rating of supervisors' desired leadership qualities was 2.9 (on the 7LQQ scale ranging from 0 to 4). On average, employees reported somewhat more restrictive than liberal workplace drinking social norms (M=2.2 on the DNS ranging from 1 to 4). Descriptive statistics for the main study variables are presented in table 3.

Bivariate correlations between all study variables (outcomes, predictors and covariates) are presented in the online supplemental table 1.

**Table 3** Descriptive statistics for the main study variables

| | Variable | |
|---|---|---|
| O | Alcohol consumption*, M (SD) | 3.21 (1.87) |
| O | Alcohol-related problems†, n (%) yes | 619 (11.50) |
| **Employment characteristics** | | |
| P | Working hours‡, M (SD) | 7.95 (1.83) |
| P | Day job, no weekends, n (%) yes | 3982 (73.90) |
| P | Evening job, n (%) yes | 22 (0.40) |
| P | Night job, n (%) yes | 84 (1.60) |
| P | Shift work, n (%) yes | 981 (18.20) |
| P | Job position, n (%) workers | 4288 (79.60) |
| P | Job size, n (%) full-time | 4407 (81.80) |
| P | Income§, M (SD) | 10.12 (6.25) |
| **Job demands and support factors** | | |
| P | Psychological job demands¶, M (SD) | 2.17 (0.43) |
| P | Job control¶, M (SD) | 3.00 (0.34) |
| P | Workplace social support¶, M (SD) | 3.19 (0.47) |
| P | Leadership qualities**, M (SD) | 2.91 (0.81) |
| **Travels and worksite factors** | | |
| P | Job travels, n (%) yes | 804 (14.90) |
| P | Working from home, n (%) yes | 374 (6.90) |
| P | Working from holiday home, n (%) yes | 161 (3.00) |
| **Workplace drinking social norms** | | |
| P | Drinking social norms††, M (SD) | 2.23 (0.53) |

n=5388.
*Alcohol Use Disorders Identification Test–Consumption (AUDIT-C), scale 0–12.
†AUDIT (all 10 items, scale 0–40), sum score ≥8.
‡Number of hours during a typical day.
§Gross annual household income in 100 000 Norwegian kroner.
¶Job Content Questionnaire, scale 1–4.
**Seven-item Leadership Qualities Questionnaire.
††Drinking Norms Scale, scale 1–4 (higher scores=more liberal norms).
M, mean; O, outcome; P, predictor; SD, Standard deviation.

### Workplace factors and alcohol consumption

Associations between workplace factors and alcohol consumption are presented in table 4.

As shown in table 4 (models 1–3), employment characteristics, job demands and support factors and travels and worksite factors explained only small proportions of the variance in alcohol consumption (1.2%–1.8%). The proportion of explained variance in consumption increased considerably (to 19.2%) when workplace drinking social norms was included in the analysis (model 4). In the fully adjusted model (model 5), also taking covariates into account, eight workplace factors were significantly associated with alcohol consumption. The highest VIF was 3.62, indicating that multicollinearity was not a concern.

Higher consumption was associated with shorter working hours, being a supervisor (vs a worker), working full-time (as opposed to part-time), higher income, higher levels of workplace social support, having a supervisor with less desired leadership qualities, working from holiday home and more liberal workplace drinking social norms. The remaining workplace factors either failed to demonstrate significant relationships with alcohol consumption altogether (evening job schedule, night job schedule, shift work schedule, job control, working from home), or failed to reach statistical significance in the fully adjusted model (day job without weekend schedule, psychological job demands, job travels).

The relative importance of the eight significant workplace factors for explaining the variance in alcohol consumption is presented in table 5.

Based on average contribution to the explained variance in alcohol consumption, drinking social norms (17.92%) was the dominant predictor, followed by job size (0.45%), working from holiday home (0.16%), job position (0.13%), leadership qualities (0.07%), working hours (0.05%), workplace social support (0.02%) and income (0.00%). In the fully adjusted regression model (table 4, model 5), a one-unit increase in drinking social norms (towards more liberal) was associated with an increase of 1.37 units on the alcohol consumption scale (ranging from 0 to 12).

### Workplace factors and alcohol-related problems

Associations between workplace factors and odds of having alcohol-related problems are presented in table 6.

As shown in table 6 (models 1–3), employment characteristics, job demands and support factors and travels and worksite factors explained only small proportions of the variance in odds of having alcohol-related problems (2.0%–2.5%). The explained variance increased considerably when workplace drinking social norms were included in the analysis (model 4, 12.5%). In the fully adjusted model (model 5, containing all workplace factors as well as covariates), only two workplace factors were significantly related with odds of having alcohol-related problems. There was no evidence of poor model fits, as indicated by Hosmer and Lemeshow tests reaching p≥0.05.

Elevated odds of alcohol-related problems was associated with shorter working hours and more liberal workplace drinking social norms. The remaining workplace factors either failed to reach statistical significance in the fully adjusted model (job size, income, job travels, working from home), or were unrelated with alcohol-related problems altogether (day job no weekend schedule, evening schedule, night schedule, shift work schedule, job position, psychological job demands, job control, workplace social support, leadership qualities, working from holiday home).

Dominance analysis (table 7) revealed that drinking social norms was the dominant predictor of alcohol-related problems by having an average contribution to the

**Table 4** Associations between workplace factors and alcohol consumption*

| | Model 1 Employment characteristics | Model 2 Job demands and support factors added | Model 3 Travels and worksite factors added | Model 4 Workplace drinking social norms added | Model 5 Fully adjusted model, including covariates† |
|---|---|---|---|---|---|
| | b (β) (p value) | b (β) (p value) | b (β) (p value) | b (β) (p value) | b (β) (p value) |
| Working hours | −0.01 (−0.00) (0.750) | 0.00 (0.00) (0.905) | −0.00 (−0.00) (0.831) | **−0.03 (−0.03) (0.029)** | **−0.03 (−0.03) (0.049)** |
| Day job, no weekends‡ | **−0.25 (−0.06) (0.022)** | **−0.28 (−0.07) (0.011)** | **−0.25 (−0.06) (0.020)** | −0.12 (−0.03) (0.231) | −0.04 (−0.01) (0.669) |
| Evening job‡ | 0.41 (0.01) (0.320) | 0.39 (0.01) (0.346) | 0.43 (0.01) (0.303) | 0.21 (0.01) (0.574) | 0.07 (0.00) (0.840) |
| Night job‡ | 0.02 (0.00) (0.920) | −0.06 (−0.00) (0.798) | −0.02 (−0.00) (0.924) | 0.12 (0.01) (0.590) | −0.04 (−0.00) (0.867) |
| Shift work‡ | 0.06 (0.01) (0.619) | 0.04 (0.01) (0.745) | 0.07 (0.02) (0.550) | 0.09 (0.02) (0.441) | −0.06 (−0.01) (0.591) |
| Job position§ | 0.11 (0.02) (0.109) | **0.14 (0.03) (0.040)** | 0.10 (0.02) (0.118) | **0.25 (0.06) (<0.001)** | **0.25 (0.06) (<0.001)** |
| Job size¶ | **0.46 (0.09) (<0.001)** | **0.46 (0.09) (<0.001)** | **0.44 (0.09) (<0.001)** | **0.31 (0.06) (<0.001)** | **0.18 (0.04) (0.006)** |
| Income** | 0.00 (0.00) (0.760) | 0.00 (0.01) (0.716) | 0.00 (0.00) (0.966) | 0.00 (0.00) (0.840) | **0.02 (0.05) (<0.001)** |
| Psychological job demands†† | – | **−0.15 (−0.03) (0.017)** | **−0.15 (−0.03) (0.016)** | **−0.13 (−0.03) (0.026)** | −0.09 (−0.02) (0.106) |
| Job control†† | – | −0.12 (−0.02) (0.144) | −0.15 (−0.03) (0.063) | −0.10 (−0.02) (0.165) | 0.02 (0.00) (0.755) |
| Workplace social support†† | – | **0.15 (0.04) (0.030)** | **0.16 (0.04) (0.023)** | **0.14 (0.03) (0.035)** | **0.13 (0.03) (0.040)** |
| Leadership qualities‡‡ | – | **−0.09 (−0.04) (0.028)** | **−0.09 (−0.04) (0.033)** | **−0.10 (−0.04) (0.005)** | **−0.10 (−0.04) (0.004)** |
| Job travels§§ | – | – | **0.25 (0.05) (0.001)** | 0.04 (0.01) (0.564) | 0.04 (0.01) (0.575) |
| Working from home§§ | – | – | −0.05 (−0.01) (0.647) | −0.18 (−0.02) (0.069) | −0.12 (−0.02) (0.216) |
| Working from holiday home§§ | – | – | **0.40 (0.04) (0.012)** | **0.42 (0.04) (0.003)** | **0.40 (0.04) (0.007)** |
| Drinking social norms¶¶ | – | – | – | **1.51 (0.42) (<0.001)** | **1.37 (0.39) (<0.001)** |
| R² | 0.012 | 0.014 | 0.018 | 0.192 | 0.225 |
| VIF range | 1.02–3.54 | 1.03–3.56 | 1.03–3.59 | 1.03–3.60 | 1.08–3.62 |

n=5388; results from multiple hierarchical linear regression analyses; b=unstandardised regression coefficient; β=standardised regression coefficient.
Bold typeface indicates statistically significant associations (p<0.05).
*Alcohol Use Disorders Identification Test-Consumption, scale 0–12, higher scores indicate higher consumption.
†Covariates=gender, age, educational attainment, living status, marital status, number of children, year of data collection.
‡Reference=other work schedules.
§Reference=worker.
¶Reference=part-time.
**Household gross annual income in 100000 Norwegian kroner.
††Job Content Questionnaire, scale 1–4, higher scores indicate higher predictor values.
‡‡Seven-item Leadership Qualities Questionnaire, scale 0–4, higher scores indicate higher level of desired leadership qualities.
§§Reference=no.
¶¶Drinking Norms Scale, scale 1–4, higher scores indicate more liberal workplace drinking norms.
VIF, variance inflation factor.

**Table 5** Relative importance of workplace factors* in explaining alcohol consumption†

| Predictor | Dominance statistic | Average contribution (pp)‡ | Ranking |
|---|---|---|---|
| Drinking social norms | 0.1792 | 17.92 | 1 |
| Job size | 0.0045 | 0.45 | 2 |
| Working from holiday home | 0.0016 | 0.16 | 3 |
| Job position | 0.0013 | 0.13 | 4 |
| Leadership qualities | 0.0007 | 0.07 | 5 |
| Working hours | 0.0005 | 0.05 | 6 |
| Workplace social support | 0.0002 | 0.02 | 7 |
| Income | 0.0000 | 0.00 | 8 |

n=5388; results from dominance analysis.
*Workplace factors that demonstrated significant associations with alcohol consumption in the fully adjusted regression model (model 5 in table 4).
†Alcohol Use Disorders Identification Test–Consumption, scale 0–12.
‡Average contribution to the $R^2$ in percentage points.

proportion of explained variance of 5.26%. In contrast, the contribution of working hours was only 0.04% (on average).

A 1-hour increase in working hours was associated with reduced odds of alcohol-related problems by a factor of 0.94. A one-unit increase on the drinking social norms scale (towards more liberal) was associated with increased odds of alcohol-related problems by a factor of 3.52.

### Summary of main results

Main results are summarised in figure 1.

Eight out of 16 workplace factors were not significantly related to either alcohol consumption or alcohol-related problems. Six workplace factors were significantly associated with alcohol consumption but not with alcohol-related problems (job position, job size, income, workplace social support, leadership qualities, working from holiday home). Two workplace factors were significantly related to both alcohol consumption and alcohol-related problems (working hours, drinking social norms). With the exception of drinking social norms (that was the dominant predictor of alcohol consumption as well as of alcohol-related problems), associations between workplace factors and the alcohol outcomes were quite weak (see tables 4 and 6) and most of the variance in employees' alcohol consumption and alcohol-related problems was explained by variables not included in this study, that is, by variables beyond the included workplace factors.

### DISCUSSION

The aim of this study was to explore whether workplace factors were associated with alcohol use in a heterogeneous sample of employees in Norway.

Drinking social norms emerged as the supremely most important predictor of both alcohol consumption and alcohol-related problems. More liberal attitudes toward the role of alcohol in workplace settings were associated with having a higher consumption as well as elevated odds of having alcohol-related problems. Drinking norms prescribe the level of consumption that is considered appropriate or tolerable in specific social contexts, and employees' drinking norms may reflect the workplace drinking culture.[26 74–76] Hence, our study supports previous research demonstrating that workplace alcohol use climate may affect employees' level and experienced consequences of alcohol consumption.[25–28] The association between workplace drinking norms and alcohol consumption can be viewed in conjunction with evidence that has demonstrated that a significant proportion of employees' overall consumption indeed does occur in job-related contexts. Studies from Norway have found that 30% of employees had consumed alcohol in job-related settings during the past 2 weeks,[77] 43% of employees' overall consumption occurred in job-related situations,[77] and employers initiated and organised more than half of the job-related situations in which employees were exposed to alcohol.[78]

Interestingly, longer work hours were associated with lower consumption but also with higher odds of having alcohol-related problems. These findings may, intuitively, stand out as somewhat contradictory, insofar that one may expect alcohol consumption to be closely associated with alcohol-related problems. We did find a considerable and statistically significant correlation between the two alcohol outcomes. However, the strength of the correlation was in the upper layer of moderate strength (see online supplemental table 1), which is far from a perfect correlation. Hence, somewhat different mechanisms may underlie the two alcohol outcomes, which has been demonstrated in earlier research. As captured by the term the alcohol harm paradox, certain populations may be exposed to a variety of health risks that interact with alcohol consumption to create synergistically detrimental effects of consumption,[79] which result in greater alcohol-related harm among individuals with specific characteristics (eg, socioeconomic variables) for each consumed alcohol unit, compared with individuals without these specific characteristics.[24] Associations between working hours and alcohol outcomes have been explained in terms of stress.[29 34 80] Shorter working hours may be related to higher consumption (as found in our study) due to involuntary part-time and/or temporary contracts generating stress that is coped with by means of higher alcohol consumption.[34] On the other hand, particularly long working hours may be associated with particularly high work loads and stress that result in coping mechanisms generating alcohol-related problems.[34 80] However,

**Table 6** Associations between workplace factors and odds of having alcohol-related problems*

| | Model 1 Employment characteristics | Model 2 Job demands and support factors added | Model 3 Travels and worksite factors added | Model 4 Workplace drinking social norms added | Model 5 Fully adjusted model, including covariates† |
|---|---|---|---|---|---|
| | OR (p value) | OR (p value) | OR (p value) | OR (p value) | OR (p value) |
| Working hours | 0.97 (0.270) | 0.98 (0.315) | 0.97 (0.247) | **0.95 (0.025)** | **0.94 (0.024)** |
| Day job, no weekends‡ | 0.74 (0.079) | 0.73 (0.074) | 0.74 (0.082) | 0.82 (0.278) | 0.97 (0.852) |
| Evening job‡ | 2.35 (0.117) | 2.30 (0.128) | 2.35 (0.119) | 1.98 (0.235) | 1.77 (0.334) |
| Night job‡ | 1.59 (0.212) | 1.48 (0.299) | 1.52 (0.268) | 1.68 (0.182) | 1.48 (0.328) |
| Shift work‡ | 1.05 (0.812) | 1.02 (0.921) | 1.04 (0.822) | 1.06 (0.761) | 0.83 (0.373) |
| Job position§ | 0.82 (0.082) | 0.83 (0.107) | 0.80 (0.067) | 0.95 (0.646) | 0.99 (0.956) |
| Job size¶ | **1.92 (<0.001)** | **1.91 (<0.001)** | **1.88 (<0.001)** | **1.68 (<0.001)** | 1.31 (0.064) |
| Income** | **1.00 (0.001)** | **1.00 (0.001)** | **1.00 (0.001)** | **1.00 (<0.001)** | 1.00 (0.764) |
| Psychological job demands†† | – | 1.02 (0.880) | 1.02 (0.869) | 1.03 (0.780) | 1.05 (0.672) |
| Job control†† | – | 0.82 (0.142) | 0.79 (0.086) | 0.82 (0.175) | 0.92 (0.565) |
| Workplace social support†† | – | 0.97 (0.764) | 0.97 (0.812) | 0.90 (0.407) | 0.88 (0.331) |
| Leadership qualities‡‡ | – | 0.94 (0.317) | 0.94 (0.329) | 0.92 (0.223) | 0.90 (0.127) |
| Job travels§§ | – | – | **1.35 (0.016)** | 1.12 (0.383) | 1.09 (0.506) |
| Working from home§§ | – | – | 0.99 (0.964) | 0.86 (0.435) | 0.91 (0.646) |
| Working from holiday home§§ | – | – | 1.09 (0.749) | 1.15 (0.599) | 1.17 (0.564) |
| Drinking social norms¶¶ | – | – | – | **4.61 (<0.001)** | **3.52 (<0.001)** |
| $R^2_{Nagelkerke}$ | 0.020 | 0.022 | 0.025 | 0.125 | 0.184 |
| Hosmer and Lemeshow test | χ=12.39, p=0.135 | χ=13.63, p=0.092 | χ=6.21, p=0.624 | χ=9.39, p=0.310 | χ=8.22, p=0.412 |

n=5388; results from multiple binary logistic regression analyses.
Bold typeface indicates statistically significant associations ($p<0.05$).
*Alcohol Use Disorders Identification Test (all 10 items), sum score 0–7 (no problems) vs 8–40 (problems).
†Covariates=gender, age, educational attainment, living status, marital status, number of children, year of data collection.
‡Reference=other work schedules.
§Reference=worker.
¶Reference=part-time.
**Household gross annual income in 100 000 Norwegian kroner.
††Job Content Questionnaire, scale 1–4, higher scores indicate higher predictor values.
‡‡Seven-item Leadership Qualities Questionnaire, scale 0–4, higher scores indicate higher level of desired leadership qualities.
§§Reference=no.
¶¶Drinking Norms Scale, scale 1–4, higher scores indicate more liberal workplace drinking norms.
OR, odds ratio.

our study did not allow causal explanations for the differential findings regarding the relationships between working hours, consumption and alcohol-related problems. It should be noted that associations found in our study were quite weak and that we, in line with earlier researchers, were unable to establish consistent knowledge regarding the relationship between working hours and alcohol outcomes among employees.[29–34]

Higher income was associated with higher alcohol consumption. This finding is consistent with earlier studies emphasising that higher socioeconomic status is related to higher levels of consumption.[23 42 43] Alcohol is a costly commodity,[44] and high-income employees have better access to alcohol than their lower-income counterparts.

Working full-time (as opposed to part-time) was associated with elevated alcohol consumption among employees. This finding contradicts earlier research from the USA that has found that part-time workers had higher weekly consumption than full-time workers.[41] Our data do not illuminate reasons for why full-time employees were more exposed to alcohol than part-time employees. One may hypothesise that this association can be interpreted in terms of the degree to which employees are

**Table 7** Relative importance of workplace factors* in explaining odds of having alcohol-related problems†

| Predictor | Dominance statistic | Average contribution (pp)‡ | Ranking |
|---|---|---|---|
| Drinking social norms | 0.0526 | 5.26 | 1 |
| Working hours | 0.0004 | 0.04 | 2 |

n=5388; results from dominance analysis.
*Workplace factors that demonstrated significant associations with odds of having alcohol-related problems in the fully adjusted regression model (model 5 in table 6).
†Alcohol Use Disorders Identification Test (all 10 items), sum score 0–7 (no problems) vs 8–40 (problems).
‡Average contribution to the $R^2$ in percentage points.

socially integrated at the workplace. Part-time employees may have a lower workplace social integration than their full-time counterparts, perhaps resulting in a lower exposure to alcohol in job-related situations. Alternatively, part-time employees may have reduced working hours due to health problems,[81] which in turn may explain why they—on a group level—consume less alcohol than full-time employees. Employees' financial situation may as well be of importance, insofar that alcohol is a costly

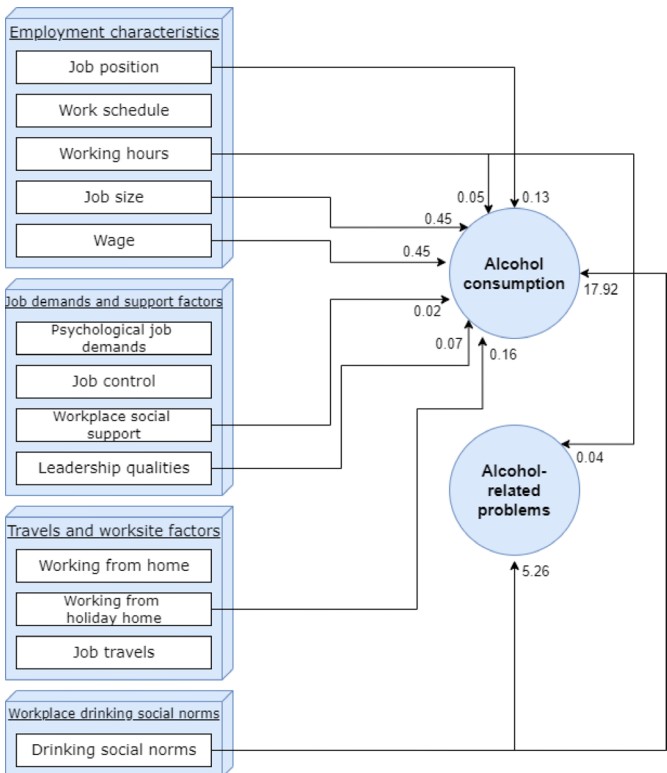

**Figure 1** Overview of main study results. Estimates reflect relative importance of predictors based on dominance analysis (average contribution of the predictor to the outcome in percentage points when analysed in conjunction with other workplace factors that are significantly associated with the outcome).

commodity,[44] and that part-time workers may find themselves in a more strained financial situation than full-time employees.

Workers consumed less alcohol than supervisors, which is in line with some earlier studies.[32 40] One may hypothesise that a supervisor position entails access and exposure to factors that may enhance alcohol consumption, such as higher income and greater work location flexibility (eg, working from holiday home). However, our data do not lend themselves to explanations for the association between job position and alcohol consumption, and several studies have failed to demonstrate such a relationship.[23 31 34 35]

Worksite factors (eg, working from home and having overall greater flexibility) and job travels have in some studies been linked with elevated alcohol consumption.[35 40 47] In our study, travels and worksite factors demonstrated mixed results. We did not find an association between alcohol consumption, working from home and job travels. However, working from holiday home was associated with elevated consumption. These findings may reflect a context dependency in the relationship between work location flexibility and alcohol use. In an earlier study exploring relationships between activity patterns and alcohol use among employees in Norway,[82] non-domestic activities were associated with higher consumption than domestic activities. Having an activity pattern characterised by activities within (rather than outside) the household was associated with lower alcohol consumption. Working from home (in a domestic setting) may constitute working in an activity setting not very compatible with alcohol use, even though working from home entails being less visible for coworkers and supervisors and less susceptible to workplace social control (factors which have been linked with elevated alcohol consumption).[9 22] Working from holiday home constitutes a non-domestic activity where work is performed in a context more similar to vacations. Vacations and holidays typically involve greater alcohol exposure, potentially resulting in higher consumption levels.[83 84]

Having supervisors with desired leadership qualities was associated with lower alcohol consumption. Supervisors who are highly rated on desired leadership qualities (problem-solving, contact-making, responsible-making, protecting, trust-creating, recognising and encouraging[69]) may be perceived as active, visible and involved with their staff and can be well positioned to set norms and exert informal social control. It has been advocated that visible and active supervisors represent protective factors for elevated alcohol use among workers,[46] and studies have demonstrated lower prevalence of drinking and hangover episodes at work among employees employed at workplaces where supervisors exert active social control.[25 50 51] As such, supervisors with desired leadership qualities may be influential in determining workplace drinking cultures. One may assume that desired leadership qualities represent a potential protective factor for elevated alcohol consumption in line with

how earlier studies have found such desired leadership to be a protective factor for mental distress[85] and physiological pain.[86 87]

In contrast to earlier studies that have found that higher levels of social support at work are associated with lower alcohol consumption among employees,[33 38 55] we found an association in the opposite direction, that is, that higher levels of support were associated with higher consumption. Social support may be conceptualised as a proxy of social integration,[88] and our measure of social support included support from both peers and supervisors. Perceived levels of social support may reflect the level of which the individual employee feels socially integrated in the workplace culture. Hence, social support may have dual and opposing effects on alcohol outcomes, depending on the workplace culture. In liberal drinking cultures, high levels of perceived support may reinforce permissive drinking norms,[89] while the opposite may be true in more restrictive workplace drinking cultures.

## Methodological considerations

The cross-sectional design of this study precludes causal inferences regarding the relationships between workplace factors and alcohol-related outcomes. Even though we attempted to minimise risk of confounding by including relevant covariates in adjusted analyses, reversed, reciprocal and/or third-variable causation cannot be ruled out. For instance, the association between job size and alcohol use may be due to job size affecting drinking pattern, to drinking pattern affecting job size, or to an unmeasured third variable (eg, health status) affecting both job size and drinking pattern. However, the aim of this study was not to unravel causal relationships but was explorative in nature. The cross-sectional design may also have played a role in explaining why observed effect sizes were quite small. Potential effects of some workplace factors on broad indicators of alcohol use and problems may be temporary and/or short lived, rendering it difficult to fully capture true effects without more robust research designs that involve longitudinal measures.

The sample was large and consisted of a heterogeneous selection of employees across work sectors, divisions, levels and geographical locations. The sample size well exceeded the a priori defined criteria,[61 62] and was thus considered statistically satisfactory (required sample size: N=435; actual sample size: N=5388). However, the response rate was quite low (27.7%). Low response rates may represent a threat to external validity by limiting generalisability of results in instances where responders systematically differ from non-responders. Study selection analyses of the WIRUS screening data indicate that the final study sample was similar to the invited sample (ie, all employees in the included companies) regarding distributions of gender and age.[90] However, compared with the entire Norwegian workforce, female, older and well-educated employees were over-represented in our study. Moreover, it must be noted that public sector employees were largely over-represented, resulting in a limited potential of generalising results across employment sectors. Probable reasons for this over-representation include that public companies in Norway, on average, tend to be larger than private companies (the project aimed to recruit companies with >100 employees), and that an economic downturn in Norway in 2014–2015 (due to falling oil prices) made it difficult for private (profit-based) companies to prioritise participation in research. There may also be important differences between the study sample and non-responders that we lack information about (unmeasured selection effects) that have biased our findings. Moreover, a considerable number of employees (n=3154) were excluded due to not responding on all study items in the survey. Comparisons between completers and non-completers on sociodemographic variables (see online supplemental table 2) demonstrated that completers were characterised by a slight over-representation of men, employees with university/college education, married employees and employees having more children.

This study was solely based on self-reported data from employees, which may involve risks of measuring bias (eg, recall bias and social desirability bias). In particular, one may assume that alcohol use may have been underestimated, insofar that studies have demonstrated a discrepancy between self-reported consumption and actual alcohol sales.[91] On the other hand, alcohol consumption and alcohol-related problems were both measured with the AUDIT, which has demonstrated psychometric properties superior to other alcohol screening instruments,[92] and has been deemed appropriate for use in Norwegian employee populations.[93] A threshold of 8 points on the AUDIT as indicative of alcohol-related problems has been found to represent a satisfactory compromise between sensitivity and specificity,[7] and is in line with previous research on non-clinical populations.[94–97] Furthermore, regarding issues of measurement, we only differentiated between workers and supervisors, that is, not taking into account how many workers supervisors were responsible for. This may have concealed some potential nuances regarding stress and workload among supervisors. We did, however, include psychological job demands as a predictor in this study, which may have captured some of these potential nuances.

## Implications

Among the workplace factors explored in this study, drinking social norms emerged as the supremely most important predictor of employee alcohol use. Hence, results from our study call attention to the importance of workplace drinking cultures to understand employees' level of alcohol consumption and odds of experiencing alcohol-related problems. Furthermore, other significant predictors of employee alcohol use support the notion of emphasising workplace drinking culture. Perceiving higher levels of social support and being full-time employed were both associated with higher consumption levels and can both be conceptualised as proxies of social integration at the workplace,

thereby indicating the importance of workplace culture. Moreover, results indicate the importance of supervisor behaviour. Supervisors consumed more alcohol than workers, and having supervisors with high levels of desired leadership qualities was associated with lower levels of consumption among employees. Efforts aimed at reducing alcohol consumption and alcohol-related problems among employees may include fostering less liberal drinking cultures at workplaces in combination with desired leadership qualities among supervisors. Raising supervisors' awareness of their influence on workplace drinking culture may be serviceable. Prosocial supervisors may stimulate workplace cultures that discourage or inhibit drinking in work-related situations and may also foster social relationships and trust that enable establishment of routines for early identification and aid for employees who may benefit from alcohol prevention interventions.

In a systematic review,[98] collaboration/teamwork and positive, accessible and fair supervisors were identified among the most pronounced factors considered as important for a healthy workplace. In another systematic review,[99] covering workplace resources to improve employee well-being and performance, social support and high-quality relationships between workers and supervisors were identified among the factors organisations should emphasise. As such, a focus on workplace factors identified in our study as potential protective factors for alcohol-related problems seems to converge with the WHO framework for healthy workplaces, that is, workplaces 'in which workers and managers collaborate to use a continual improvement process to protect and promote health, safety and well-being of all workers and the sustainability of the workplace by considering' important aspects such as the psychosocial work environment, the organisation of work, and the workplace culture[100] (p. 16).

Evidence on associations between workplace factors and employee alcohol use still stands out as complex and nuanced, and this study showed that only marginal proportions of the variance in alcohol consumption and alcohol-related problems were explained by employment characteristics, job demands and support factors and travels and worksite factors. Hence, further research is warranted. Results from this study suggest the application and further development of interventions, for example, Workplace Health Promotion Programs, targeting workplace drinking culture and leadership. Effects and implementation of such interventions should be explored by means of robust research designs, such as prospective cohort studies and cluster randomised controlled trials. Future research may also benefit from explorations of potential interactions between individual and workplace factors, including mediators and moderators, which were beyond the scope of our study.

## CONCLUSIONS

Knowledge about workplace factors associated with employee alcohol use is important as an aid in determining the extent to which employers should make workplace-based alcohol prevention interventions an overall priority. This study, conducted in a large and heterogeneous sample of employees in Norway, points to the importance of drinking social norms, workplace drinking culture and leadership.

**Author affiliations**
[1]Department of Rehabilitation Science and Health Technology, Faculty of Health Sciences, Oslo Metropolitan University, Oslo, Norway
[2]Department of Public Health, Faculty of Health Sciences, University of Stavanger, Stavanger, Norway
[3]Department of Health Promotion, Norwegian Institute of Public Health, Bergen, Norway
[4]Center for Alcohol & Drug Research, Stavanger University Hospital, Stavanger, Norway
[5]Department of Health and Nursing Sciences, Faculty of Social and Health Sciences, Inland Norway University of Applied Sciences, Elverum, Norway
[6]Department of Health, Faculty of Health Studies, VID Specialized University, Stavanger, Norway

**Contributors** RWA is the principal investigator (PI) and project manager (PM) for the WIRUS Project (Workplace Interventions preventing Risky alcohol Use and Sick leave). This study was designed by MMT and RWA. MMT analysed the data and drafted the manuscript. JCS, TB, LSS, RWA and MMT provided scientific input to the different drafts and provided data interpretation. All authors made critical revisions and provided intellectual content to the manuscript, approved the final version to be published and agreed to be accountable for all aspects of this work. RWA is the guarantor of this article.

**Funding** This work was supported by the Norwegian Directorate of Health (grant number: n/a) and the Research Council of Norway (grant number: 260640).

**Competing interests** None declared.

**Patient and public involvement** Patients and/or the public were involved in the design, or conduct, or reporting, or dissemination plans of this research. Refer to the Methods section for further details.

**Patient consent for publication** Not required.

**Ethics approval** This study involves human participants and was approved by the Regional Committee for Medical and Health Research in Norway (REK; approval no. 2014/647). Participants gave informed consent to participate in the study before taking part.

**Provenance and peer review** Not commissioned; externally peer reviewed.

**Data availability statement** Data are available upon reasonable request. De-identified data from the WIRUS screening study are available from the corresponding author on reasonable request.

**ORCID iD**
Mikkel Magnus Thørrisen http://orcid.org/0000-0001-9869-6541

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
