## [Reviewer comments · BMJ Open]

ARTICLE DETAILS

TITLE (PROVISIONAL)	Are workplace factors associated with employee alcohol use? The WIRUS cross-sectional study
AUTHORS	Thørrisen, Mikkel; Skogen, Jens; Bonsaksen, Tore; Skarpaas, Lisebet; Aas, Randi

VERSION 1 – REVIEW

REVIEWER	Peter Bamberger Tel Aviv University Coller School of Management, Department of Organization Behavior
REVIEW RETURNED	11-Jun-2022

GENERAL COMMENTS	This manuscript uses data from over 6000 workers and supervisors employed in 22 Norwegian enterprises to examine the degree to which work-related factors explain workforce alcohol consumption and drinking problems. The specific risk factors examined relate to employment characteristics (e.g., hours of work, work shift, and the nature of the position), as well as the psycho-social work context (e.g., leadership qualities, work demands, and social support). Study strengths include a large and diverse sample, the application of validated measures of alcohol use and misuse, and a strong and well-grounded discussion section. Nevertheless, there are a number of steps that can be taken to strengthen the potential contribution of this paper. 1. Positioning: The paper is largely grounded on the proposition that findings regarding work-related risk factors for workforce alcohol use and misuse are inconsistent. That may in fact be the case for some of the variables examined in this study (particularly those relating to employment characteristics), but findings regarding psychosocial risk factors are far less equivocal. These risk factors can be grouped into 3 main categories, namely the workplace use climate (capturing more permissive vs. restrictive alcohol-related norms), social control, and stress. As summarized by Frone and Bamberger (In Press), findings regarding the impact of injunctive and descriptive norms (key use climate variables) are quite consistent. Findings regarding social control and stress are indeed more equivocal when focusing on direct effects. However, when boundary conditions (interactions with such individual difference variables as positive alcohol expectancies or neuroticism) are taken into consideration, here too there is a reasonable degree of consistency. Hence, it may be preferable to instead argue that prior findings suggest highly nuanced effects of workplace factors on alcohol use and misuse. 2. Selection of risk factors: Prior research suggests that employment-related characteristics (other than occupation/vocation),
--

even if linked to alcohol outcomes, tend to have limited effect sizes. Accordingly, further justification for focusing on these factors would be helpful. One way of better justifying the selection of risk factors might be to group them into the three key categories noted above. More specifically, employment characteristics such as job travels, working from home/holiday home) relate to social control (i.e., being under less direct monitoring by supervisors and peers), whereas job demands, control and support are the three components of the Karasek model of occupational stress. If available, variables related to the workplace use climate should be examined as well in order to offer a more comprehensive analysis. Given that leadership is measured in terms of supportive supervision in the current study (as opposed to abusive supervision – see Bamberger & Bacharach, 2006), it may best be framed as an additional stress mitigation factor (similar to job control or social support).

3. Model: A major weakness of the study is its focus on direct effects. While demographic covariates are taken into account in one of the study models, in no case are the work-related risks tested competitively (i.e., in the context of multi-variate models including other risk factors). If variables are grouped into broader categories such as workplace stressors and social control factors, a series of hierarchical models might be useful. Moreover, despite consistent findings that certain work-related factors heighten vulnerabilities or mitigate the effects of other work-related factors (see Frone 2013; Frone & Bamberger, In Press), no moderation (i.e., interaction) effects are either proposed or tested. Given prior findings regarding the nuanced nature of the effects of work-related risk factors, and particularly in the context of such a large sample size, power fails to serve as a reasonable justification for focusing on simple, direct effects only.

4. Empirics:

a. Please expand Table 3 (descriptive stats) to include a correlation matrix. This will help the reader understand the potential empirical overlaps in the variables

b. Please consider expanding Table 4 and the discussion of it to include effects for overall AUDIT-C (crude effects aren't necessary; give only effects when controlling for covariates), as well as for modal alcohol consumption (AUDIT 1 x AUDIT 2), and for frequency of heavy drinking/binging (AUDIT 3).

c. Given the cross-sectional nature of the analysis and the resulting high probability of CMV, controls for (a) social desirability, and (b) positive or negative affect should be included. If these variables are not available, consider estimating the variance inflation factor and providing it as evidence that CMV is of limited risk.

5. Limitations not noted but of significance: The authors do a good job identifying the limitations of their study, but several concerns are not mentioned.

a. One has to do with the fact that nearly 80% of their sample is employed in public administration/services. If in fact, 80% of employees are engaged in the public sector, this should be made more clear, and issues of external validity should be noted in the limitations section.

b. Another has to do with the very low effect sizes (substantially lower than that found in other studies). Further discussion would help to offer insight into why effect sizes here were so much lower than in other studies. For example, reference may be made to the cross-sectional nature of the study, and reliance on broad, modal

	indicators of alcohol use and misuse. Studies indicate that the effects of workplace factors may be short-lived, such that heightened demands or poor leadership on a given day may have heightened effects on the same or next day. The current study design is unable to capture such effects. c. A third has to do with the measure of social support. To the extent that peer support revolves around alcohol consumption, this factor may have dual, but opposing effects. Namely, while it may mitigate stress, social support may also reinforce permissive drinking norms (See Bacharach et al., 2007). References Bacharach, S. B. et al. (2007). Retirement, social support, and drinking behavior: a cohort analysis of males with a baseline history of problem drinking. Journal of Drug Issues, 37(3), 525-548. Bamberger, P. A., & Bacharach, S. B. (2006). Abusive supervision and subordinate problem drinking: Taking resistance, stress and subordinate personality into account. Human Relations, 59(6), 723-752. Frone, M. R. (2013). Alcohol and illicit drug use in the workforce and workplace. American Psychological Association. Frone, M. R. & Bamberger, P.A. (In Press). Alcohol and illicit drug involvement in the workforce and workplace. Chapter 18 in J. Quick, G. Fisher, M. Ford, and L. Tetrick (Eds.), Handbook of Occupational Health Psychology. American Psychological Association.
--	--

REVIEWER	Trudy Sullivan Otago University, Preventive and Social Medicine
REVIEW RETURNED	14-Jun-2022

GENERAL COMMENTS	Review: Associations between workplace factors and employee alcohol use: the WIRUS Screening study I enjoyed reading this paper. It is well written and interesting including extensive use of the literature. The methodological section clearly outlines the limitations which strengthens the paper. My comments are relatively minor and mostly relate to providing a bit more context/clarification. I've also noted some minor edits which you may find useful. I'm not sure about the first sentence in the Abstract. Yes – reducing harmful alcohol use is pivotal in promoting good health but I'm not sure it is a primary objective of your research. The objectives are exploring the associations between workplace factors and alcohol use as outlined in the next two sentences. The impact of alcohol on health is certainly part of the background and appropriately included in the introduction. Data collection and sample: possibly it's because I'm unfamiliar with the Norwegian context but a bit more context in some instances would have been helpful for me. For example, I have not come across the term "pink" colour worker before though I could guess what it meant. Why is the age range to 72 years? Is that the retirement age in Norway? It seems that mainly government departments are associated with occupational health service units. Perhaps that is why the sample,
--

	although generally representative of the companies involved, is not so representative of the general population (as noted in the methodological considerations). Is there a reason why these types of companies were recruited? I think this study is part of a larger one and I imagine it would take some time to collect data from such an extensive sample but I wondered why the recruitment period was five years which is quite a long time. Was this taken into account in the analysis, i.e. a lot can change in the general environment (employment opportunities, attitudes towards alcohol etc) over such a time period which may influence responses. Why was income/salary not collected? Measures: Did you consider using type of workplace in the analysis? Or type of worker, e.g. blue, white, pink in addition to employee/supervisor? How is a supervisor defined? Did you collect data regarding supervision responsibilities, e.g. how many people were supervisors responsible for? Possibly the stress/workload associated with a larger supervisory role will be captured in “psychological job demands”. Were the same survey questions used for workers and supervisors, which would assume that supervisors have their own supervisors? Figure 1. The figure implies that “supervisors’ leadership qualities” and “workplace social support” account for 100% of alcohol-related problems when in fact they account for 0.3% of the variance. The same applies with the factors influencing alcohol consumption. Though it has “(based on dominance analysis)” in the title, the Figure should be standalone and therefore this should be made clear. Minor edits: Abstract conclusion: “... of these factors was limited.” Introduction: When you say that alcohol is associated with detrimental outcomes such as disease... etc perhaps say “health outcomes” as there are other harmful outcomes from alcohol use such as increased crime, traffic accidents etc. DALYs could be abbreviated in the first sentence rather than the second. I wonder too if the last sentence of the first paragraph could be split with the second part written first, i.e. “Risky drinking has been conceptualised...problems. Reducing harmful alcohol use represents a keystone in sustainable development of health.” Page 6, line 19: “...laissez-faire leadership) was not related...” Page 11, line 13: scored should be scores Page 17, line 5: wither? Page 19, line 16: parenthesis missing after “consumption” Page 20, line 6: parenthesis missing after “encouraging” Page 23, line 3: raise should be raising
--	--

VERSION 1 – AUTHOR RESPONSE

Reviewer #1: This manuscript uses data from over 6000 workers and supervisors employed in 22 Norwegian enterprises to examine the degree to which work-related factors explain workforce alcohol consumption and drinking problems. The specific risk factors examined relate to employment characteristics (e.g., hours of work, work shift, and the nature of the position), as well as the psycho-social work context (e.g., leadership qualities, work demands, and social support). Study strengths include a large and diverse sample, the application of validated measures of alcohol use and misuse, and a strong and well-grounded discussion section. Nevertheless, there are a number of steps that can be taken to strengthen the potential contribution of this paper.

Authors: We appreciate the reviewer's thorough reading of the manuscript, as well as insightful and constructive suggestions. We have tried our best to revise the manuscript in accordance with the reviewer's comments.

Reviewer #1: 1. Positioning: The paper is largely grounded on the proposition that findings regarding work-related risk factors for workforce alcohol use and misuse are inconsistent. That may in fact be the case for some of the variables examined in this study (particularly those relating to employment characteristics), but findings regarding psychosocial risk factors are far less equivocal. These risk factors can be grouped into 3 main categories, namely the workplace use climate (capturing more permissive vs. restrictive alcohol-related norms), social control, and stress. As summarized by Frone and Bamberger (In Press), findings regarding the impact of injunctive and descriptive norms (key use climate variables) are quite consistent. Findings regarding social control and stress are indeed more equivocal when focusing on direct effects. However, when boundary conditions (interactions with such individual difference variables as positive alcohol expectancies or neuroticism) are taken into consideration, here too there is a reasonable degree of consistency. Hence, it may be preferable to instead argue that prior findings suggest highly nuanced effects of workplace factors on alcohol use and misuse.

Authors: We agree that it is more correct to characterise the literature as "nuanced" rather than inconsistent, particularly with regard to psychosocial factors. Moreover, we agree that it is useful to categorise psychosocial factors into three main categories. Throughout the manuscript, we now conceptualise workplace factors in four broad categories, which are initially described on p. 4 (lines 10-19): "Workplace factors may, on the one hand, refer to employment characteristics, i.e., to more general workplace factors conceptualised in the form of conditions and terms defining the work situation (e.g., working hours, work schedule, job position, job size and income). On the other hand, the workplace can be understood as a psycho-socio-cultural setting that involves job demands and support factors (which are related to occupational stress, e.g., psychological job demands, job control, workplace social support and supervisors' perceived leadership qualities), travels and worksite factors (related to degree of workplace social control, e.g., job travels, working from home and working from holiday home), and workplace drinking social norms (which reflect the workplace alcohol use climate, e.g., workplace drinking culture and drinking social norms)."

Reviewer #1: 2. Selection of risk factors: Prior research suggests that employment-related characteristics (other than occupation/vocation), even if linked to alcohol outcomes, tend to have limited effect sizes. Accordingly, further justification for focusing on these factors would be helpful. One way of better justifying the selection of risk factors might be to group them into the three key categories noted above. More specifically, employment characteristics such as job travels, working from home/holiday home) relate to social control (i.e., being under less direct monitoring by supervisors and peers), whereas job demands, control and support are the three components of the Karasek model of occupational stress. If available, variables related to the workplace use climate should be examined as well in order to offer a more comprehensive analysis. Given that leadership is measured in terms of supportive supervision in the current study (as opposed to abusive supervision – see Bamberger & Bacharach, 2006), it may best be framed as an additional stress mitigation factor (similar to job control or social support).

Authors: We have now grouped the workplace factors into the four broad categories: (i) general employment characteristics, (ii) job demands and support factors, (iii) travels and worksite factors and (iv) workplace drinking social norms. This categorisation is introduced in the Introduction and utilised consistently throughout the manuscript. Based on the reviewer's suggestion, we have included drinking social norms as an additional factor in order to capture variables related to the workplace

alcohol use climate. This factor is described and analysed in line with the other workplace factors throughout the manuscript. As suggested, leadership qualities are now categorised as a job demands and support factor (related to occupational stress). Inclusion of drinking social norms (and also income as suggested by reviewer #2) made it necessary to re-run all analyses and re-write all manuscript sections. Hence, all sections of the manuscript are revised.

Reviewer #1: 3. Model: A major weakness of the study is its focus on direct effects. While demographic covariates are taken into account in one of the study models, in no case are the work-related risks tested competitively (i.e., in the context of multi-variate models including other risk factors). If variables are grouped into broader categories such as workplace stressors and social control factors, a series of hierarchical models might be useful. Moreover, despite consistent findings that certain work-related factors heighten vulnerabilities or mitigate the effects of other work-related factors (see Frone 2013; Frone & Bamberger, In Press), no moderation (i.e., interaction) effects are either proposed or tested. Given prior findings regarding the nuanced nature of the effects of work-related risk factors, and particularly in the context of such a large sample size, power fails to serve as a reasonable justification for focusing on simple, direct effects only.

Authors: We appreciate the feedback, and we agree that the focus on direct effects constituted a weakness of the study. We have re-run all analyses. Instead of focusing on direct of effects, we have now conducted multiple hierarchical regression analyses (linear and logistic) where predictors were entered sequentially in five steps. Predictors were included based on the revised categorisation of workplace factors, i.e., employment characteristics in step 1, job demands and support factors in step 2, travels and worksite factors in step 3, workplace drinking social norms in step 4, and sociodemographic covariates in step 5. The revised analytical procedure is described in the Methods section (p. 13, lines 10-16 + lines 18-23). Main results are presented in revised Table 4 (p. 17) and Table 6 (p. 20). Instead of a "manual" approach to dominance analysis (with results originally reported in Supplementary tables), we have re-run dominance analyses in Stata and reported the results (more user-friendly) in Table 5 (p. 18-19) and Table 7 (p. 21). We agree that the complexity of the relationships between workplace factors and alcohol outcomes makes it interesting to explore moderation and mediation effects. However, we believe this resides beyond the scope of our current study, which aims to explore more "higher-order" relationships between the workplace and alcohol. Explorations of moderation/mediation should, in our opinion, be based on testing of specific hypotheses, which we believe would have made our already comprehensive article too comprehensive. We have, however, specified the following in our discussion of implications for future research: "Future research may also benefit from explorations of potential interactions between individual and workplace factors, including mediators and moderators, which were beyond the scope of our study" (p. 30, lines 19-21).

Reviewer #1: 4. Empirics: a. Please expand Table 3 (descriptive stats) to include a correlation matrix. This will help the reader understand the potential empirical overlaps in the variables.

Authors: We have included a correlation matrix presenting bivariate correlations between all study variables (outcomes, predictors and covariates) in the Supplementary File (Table S1). The study consists of a total of 25 variables. Inclusion of bivariate correlations in Table 3 (descriptive statistics) would, in our opinion, have made the table too comprehensive.

Reviewer #1: b. Please consider expanding Table 4 and the discussion of it to include effects for overall AUDIT-C (crude effects aren't necessary; give only effects when controlling for covariates), as

well as for modal alcohol consumption (AUDIT 1 x AUDIT 2), and for frequency of heavy drinking/binging (AUDIT 3).

Authors: We have considered expanding Table 4 and the discussion of it as suggested. In light of our revised analytical strategy (which includes five-step hierarchical linear and logistic regression analyses, as well as corresponding dominance analyses for each outcome) we believe an inclusion of two additional outcomes (modal consumption + binge frequency) would result in a too comprehensive article.

Reviewer #1: c. Given the cross-sectional nature of the analysis and the resulting high probability of CMV, controls for (a) social desirability, and (b) positive or negative affect should be included. If these variables are not available, consider estimating the variance inflation factor and providing it as evidence that CMV is of limited risk.

Authors: Unfortunately, our data do not allow controlling for social desirability and positive/negative affect. We do, however, agree that it is serviceable to test for potential CMV. Therefore, we have calculated variance inflation factors (VIFs) for all predictors/covariates in the hierarchical regression analysis. The VIF ranges are reported in Table 4 (p. 17) and commented in-text on p. 18 (lines 6-7): "The highest VIF was 3.62, indicating that multicollinearity was not a concern". The inclusion of VIFs is described in the Methods section (p. 13, lines 15-16): "Potential multicollinearity was explored using the variance inflation factor (VIF). Multicollinearity was deemed a concern if VIFs exceeded 10 [69]". Moreover, we have calculated and reported goodness of fit for the hierarchical logistic regression models in terms of Hosmer and Lemeshow tests, described in the Methods section (p. 13, lines 20-21): "Goodness of fit was explored by means of Hosmer and Lemeshow tests, where poor model fit was indicated by $p < .05$ ". Results are reported in Table 6 (p. 20) and commented in-text on p. 21 (lines 7-8): "There was no evidence of poor model fits, as indicated by Hosmer and Lemeshow tests reaching $p \geq .05$ ".

Reviewer #1: 5. Limitations not noted but of significance: The authors do a good job identifying the limitations of their study, but several concerns are not mentioned. a. One has to do with the fact that nearly 80% of their sample is employed in public administration/services. If in fact, 80% of employees are engaged in the public sector, this should be made more clear, and issues of external validity should be noted in the limitations section.

Authors: We appreciate the feedback and suggestions for further discussion of important limitations. Limitations associated with the overrepresentation of public sector employees are now discussed on p. 28, lines 4-10: "Moreover, it must be noted that public sector employees were largely overrepresented, resulting in a limited potential of generalising results across employment sectors. Probable reasons for this overrepresentation include that public companies in Norway, on average, tend to be larger than private companies (the project aimed to recruit companies with >100 employees), and that an economic downturn in Norway in 2014-2015 (due to falling oil prices) made it difficult for private (profit-based) companies to prioritise participation in research".

Reviewer #1: b. Another has to do with the very low effect sizes (substantially lower than that found in other studies). Further discussion would help to offer insight into why effect sizes here were so much lower than in other studies. For example, reference may be made to the cross-sectional nature of the study, and reliance on broad, modal indicators of alcohol use and misuse. Studies indicate that the effects of workplace factors may be short-lived, such that heightened demands or poor leadership on

a given day may have heightened effects on the same or next day. The current study design is unable to capture such effects.

Authors: We agree that this issue should have been discussed more thoroughly in the original manuscript. We have now raised this issue in our discussion of methodological considerations on p. 27 (lines 14-18): "The cross-sectional design may also have played a role in explaining why observed effects sizes were quite small. Potential effects of some workplace factors on broad indicators of alcohol use and problems may be temporary and/or short-lived, rendering it difficult to fully capture true effects without more robust research designs that involve longitudinal measures."

Reviewer #1: c. A third has to do with the measure of social support. To the extent that peer support revolves around alcohol consumption, this factor may have dual, but opposing effects. Namely, while it may mitigate stress, social support may also reinforce permissive drinking norms (See Bacharach et al., 2007).

Authors: We appreciate this feedback, and we agree that potential dual effects of social support deserve a more thorough discussion. In light of our revised results (where social support is positively associated with alcohol consumption), we have revised our discussion of this finding on p. 27 (lines 20-25) and p. 28 (lines 1-4).

PART C: RESPONSES TO REVIEWER #2

Reviewer #2: I enjoyed reading this paper. It is well written and interesting including extensive use of the literature. The methodological section clearly outlines the limitations which strengthens the paper. My comments are relatively minor and mostly relate to providing a bit more context/clarification. I've also noted some minor edits which you may find useful.

Authors: We appreciate the reviewer's thorough reading of the manuscript, as well as insightful and constructive suggestions. We have tried our best to revise the manuscript in accordance with the reviewer's comments.

Reviewer #2: I'm not sure about the first sentence in the Abstract. Yes – reducing harmful alcohol use is pivotal in promoting good health but I'm not sure it is a primary objective of your research. The objectives are exploring the associations between workplace factors and alcohol use as outlined in the next two sentences. The impact of alcohol on health is certainly part of the background and appropriately included in the introduction.

Authors: We agree. The primary objective of the study was exploration of associations between workplace factors and alcohol use. As suggested, we have removed the first sentence in the Abstract.

Reviewer #2: Data collection and sample: possibly it's because I'm unfamiliar with the Norwegian context but a bit more context in some instances would have been helpful for me. For example, I have not come across the term "pink" colour worker before though I could guess what it meant. Why is the age range to 72 years? Is that the retirement age in Norway?

Authors: In the Methods section we have included more thorough descriptions of the two inclusion criteria in question on p. 8 (line 25) and p. 9 (lines 1-7): "Our definition of employees (criteria ii) included white- (supervisors, (semi)professional roles, business owners), blue- (labourers and skilled

trade roles) and pink-collar employed (hospitality, retail, care and administration roles) [57]. Inclusion of employees aged 16-72 (criteria i) was chosen because 16 constitutes the age when individuals in Norway have typically completed mandatory primary and secondary school and are eligible for full-time work (or further education), and because employment by Norwegian law may be terminated by the employer at age 72 (even though 67 constitutes the general age of retirement) [58].

Reviewer #2: It seems that mainly government departments are associated with occupational health service units. Perhaps that is why the sample, although generally representative of the companies involved, is not so representative of the general population (as noted in the methodological considerations). Is there a reason why these types of companies were recruited?

Authors: Probable reasons for the overrepresentation of public sector companies are now discussed in the Methodological considerations section on p. 28, lines 4-10: "Moreover, it must be noted that public sector employees were largely overrepresented, resulting in a limited potential of generalising results across employment sectors. Probable reasons for this overrepresentation include that public companies in Norway, on average, tend to be larger than private companies (the project aimed to recruit companies with >100 employees), and that an economic downturn in Norway in 2014-2015 (due to falling oil prices) made it difficult for private (profit-based) companies to prioritise participation in research".

Reviewer #2: I think this study is part of a larger one and I imagine it would take some time to collect data from such an extensive sample but I wondered why the recruitment period was five years which is quite a long time. Was this taken into account in the analysis, i.e. a lot can change in the general environment (employment opportunities, attitudes towards alcohol etc) over such a time period which may influence responses.

Authors: It is correct that this cross-sectional study is part of a larger project, which is the reason for why companies were recruited over a period of approx. five years. We agree that true temporal changes may potentially have biased our results. We have re-run all analyses and included year of data collection as a covariate, as described on p. 12 (line 13) and p. 13 (lines 1-2): "Due to the cross-sectional data in this study being collected over a period of approximately five years, year of data collection (2015; 2016; 2017; 2018; 2019) was included as an additional covariate". The results were largely similar.

Reviewer #2: Why was income/salary not collected?

Authors: Data on household gross annual income in Norwegian Kroner were collected. Originally, we decided to not include income in this study due to quite a lot of missing data on this variable. However, we do agree that income constitutes an important workplace factor. We have reconsidered our decision based on the reviewer's comment and included income as a workplace factor (among the other general employment characteristics). This decision resulted in a reduced sample size (from 6164 to 5388). However, we believe that the final sample size is statistically satisfactory.

Reviewer #2: Measures: Did you consider using type of workplace in the analysis? Or type of worker, e.g. blue, white, pink in addition to employee/supervisor? How is a supervisor defined? Did you collect data regarding supervision responsibilities, e.g. how many people were supervisors responsible for? Possibly the stress/workload associated with a larger supervisory role will be captured in

“psychological job demands”. Were the same survey questions used for workers and supervisors, which would assume that supervisors have their own supervisors?

Authors: We appreciate the feedback. We did consider including type of workplace/work (e.g., work division, blue/white/pink-collar) in our analyses, but we concluded that our data did not hold satisfactory quality in this regard. We were able to make superordinate distinctions between work divisions and employment sectors (as reported in Table 1, presenting characteristics of the study sample). However, we deemed these distinctions as not meaningful enough to include them as predictors in our analyses. In the survey, respondents were able to define themselves as "worker" or "supervisor". The same survey questions were used for workers and supervisors, and it was assumed that supervisors had their own supervisors.

Reviewer #2: Figure 1. The figure implies that “supervisors’ leadership qualities” and “workplace social support” account for 100% of alcohol-related problems when in fact they account for 0.3% of the variance. The same applies with the factors influencing alcohol consumption. Though it has “(based on dominance analysis)” in the title, the Figure should be standalone and therefore this should be made clear.

Authors: Dominance analyses have been re-run and Figure 1 has been revised. In order to clarify the meaning of percentages reported in the figure, we have revised the figure legend on p. 38 (lines 37-38) and p. 39 (lines 1-2): "Figure 1. Overview of main study results. Percentages reflect relative importance of predictors based on dominance analysis (average contribution of the predictor to the outcome in percentage points when analysed in conjunction with other workplace factors that is significantly associated with the outcome)".

Reviewer #2: Abstract conclusion: “... of these factors was limited.”

Authors: We have revised accordingly (p. 3, line 1).

Reviewer #2: Introduction: When you say that alcohol is associated with detrimental outcomes such as disease... etc perhaps say “health outcomes” as there are other harmful outcomes from alcohol use such as increased crime, traffic accidents etc.

Authors: We have revised accordingly (p. 3, line 16).

Reviewer #2: DALYs could be abbreviated in the first sentence rather than the second. I wonder too if the last sentence of the first paragraph could be split with the second part written first, i.e. “Risky drinking has been conceptualised...problems. Reducing harmful alcohol use represents a keystone in sustainable development of health.”

Authors: DALY is now abbreviated in the first sentence (p. 3, line 17). The last sentence of the first paragraph has been split and the order has been reversed, as suggested (p. 3, lines 19-22).

Reviewer #2: Page 6, line 19: “...laissez-faire leadership) was not related...”

Authors: We have revised accordingly (p. 6, line 20).

Reviewer #2: Page 11, line 13: scored should be scores

Authors: We have revised accordingly (p. 11, line 15).

Reviewer #2: Page 17, line 5: wither?

Authors: "wither" has been replaced with "either", which was the intended word (p. 21, line 10).

Reviewer #2: Page 19, line 16: parenthesis missing after "consumption"

Authors: We have revised accordingly (p. 26, line 3).

Reviewer #2: Page 20, line 6: parenthesis missing after "encouraging"

Authors: The parenthesis is intended to follow the reference (in order to clarify that the reference refers the content presented within the parentheses), as reported in the original manuscript.

Reviewer #2: Page 23, line 3: raise should be raising

Authors: We have revised accordingly (p. 29, line 18).

VERSION 2 – REVIEW

REVIEWER	Peter Bamberger Tel Aviv University Coler School of Management, Department of Organization Behavior
REVIEW RETURNED	18-Aug-2022

GENERAL COMMENTS	The authors did an excellent job of responding to my earlier comments.
--

REVIEWER	Trudy Sullivan Otago University, Preventive and Social Medicine
REVIEW RETURNED	15-Aug-2022

GENERAL COMMENTS	Thank you for addressing my comments. I have a few small queries in regard to your responses which you might like to consider. 1. I queried the age range of 16-72, specifically in regard to the upper age limit. I don't think the lower age limit needs to be explained (though it's fine to keep in) but it does raise a point - isn't the legal age of alcohol consumption in Norway 18 years? Therefore should participants aged 16-17 be included? Or maybe this should be noted?2. You didn't specifically answer my question about supervision responsibilities. It appears that participants identified themselves as either a worker or supervisor but for supervisors, there were not additional questions relating to their role, i.e. how many workers they were responsible for. It may be that those with a larger supervisory
--

	role could have greater stress/workload. Perhaps this could be noted even though it can't be analysed. 3. If the title/legend for Figure 1 (which has now been changed) is too long perhaps the additional information in parentheses could be noted below the Figure.
--	--

VERSION 2 – AUTHOR RESPONSE

Reviewer #1: The authors did an excellent job of responding to my earlier comments.

Authors: We appreciate the positive feedback from reviewer #1.

PART B: RESPONSE TO REVIEWER #2

Reviewer #2: Thank you for addressing my comments. I have a few small queries in regard to your responses which you might like to consider.

Authors: We appreciate the positive feedback from reviewer #2. We have revised the manuscript, based on reviewers #2 comments.

Reviewer #2: 1. I queried the age range of 16-72, specifically in regard to the upper age limit. I don't think the lower age limit needs to be explained (though it's fine to keep in) but it does raise a point - isn't the legal age of alcohol consumption in Norway 18 years? Therefore should participants aged 16-17 be included? Or maybe this should be noted?

Authors: It is correct that Norwegian legislation contains alcohol-related age limits of 18 years (beverages with <22% alcohol content) and 20 years (beverages with ≥22% alcohol content). However, these limits explicitly relate to retail, serving and supply, and not to consumption per se. Moreover, research has consistently shown that adolescents in Norway do indeed consume alcohol before reaching the age of 18. For instance, a recent national study indicated that 5-6% of adolescents aged 13-15 had consumed alcohol at least on a monthly basis, while the corresponding proportion for adolescents aged 16-18 was 37-43% (Bakken, 2022). In order to clarify this issue, we have added the following in the Methods section (Data collection and sample; p. 9, lines 7-9): "Although Norwegian legislation forbids retail, serving and supply of alcohol to anyone under the age of 18,[59] alcohol is regularly consumed by adolescents.[60] Hence, it was deemed appropriate to include respondents aged 16 and 17 in the study".

Reference: Bakken, A. (2022). Ungdata 2022. Nasjonale resultater [Ungdata 2022. National results]. OsloMet – Oslo Metropolitan University.

Reviewer #2: 2. You didn't specifically answer my question about supervision responsibilities. It appears that participants identified themselves as either a worker or supervisor but for supervisors, there were not additional questions relating to their role, i.e. how many workers they were responsible for. It may be that those with a larger supervisory role could have greater stress/workload. Perhaps this could be noted even though it can't be analysed.

Authors: We apologise for not addressing this issue in our revision. It is correct that we, in this study, differentiated solely between workers and supervisors. This may, we agree, have concealed some potential nuances regarding stress and workload among supervisors. We have added the following sentences at the end of the Methodological considerations section (p. 29, lines 2-6): "Furthermore, regarding issues of measurement, we only differentiated between workers and supervisors, i.e., not taking into account how many workers supervisors were responsible for. This may have concealed some potential nuances regarding stress and workload among supervisors. We did, however, include psychological job demands as a predictor in this study, which may have captured some of these potential nuances".

Reviewer #2: 3. If the title/legend for Figure 1 (which has now been changed) is too long perhaps the additional information in parentheses could be noted below the Figure.

Authors: Unfortunately, we have not been able to find information about word limits for figure legends. The legend for Figure 1 currently consists of 41 words. We believe this to be in line with other articles published in BMJ Open. For instance, the legend for Figure 3 in <https://doi.org/10.1136/bmjopen-2022-062446> consists of 53 words, while the legend for Figure 1 in <https://doi.org/10.1136/bmjopen-2021-060458> consists of 89 words. Therefore, we have not made any changes in our legend for Figure 1.